# The Exploitation of Lysosomes in Cancer Therapy with Graphene-Based Nanomaterials

**DOI:** 10.3390/pharmaceutics15071846

**Published:** 2023-06-28

**Authors:** Biljana Ristic, Mihajlo Bosnjak, Maja Misirkic Marjanovic, Danijela Stevanovic, Kristina Janjetovic, Ljubica Harhaji-Trajkovic

**Affiliations:** 1Institute of Microbiology and Immunology, Faculty of Medicine, University of Belgrade, Dr. Subotića 1, 11000 Belgrade, Serbia; 2Department of Neurophysiology, Institute for Biological Research “Siniša Stanković”—National Institute of Republic of Serbia, University of Belgrade, Despot Stefan Blvd. 142, 11000 Belgrade, Serbia

**Keywords:** graphene-based nanomaterials, graphene-based drug delivery systems, lysosomes, cancer, endosomal/lysosomal escape, lysosomal cell death

## Abstract

Graphene-based nanomaterials (GNMs), including graphene, graphene oxide, reduced graphene oxide, and graphene quantum dots, may have direct anticancer activity or be used as nanocarriers for antitumor drugs. GNMs usually enter tumor cells by endocytosis and can accumulate in lysosomes. This accumulation prevents drugs bound to GNMs from reaching their targets, suppressing their anticancer effects. A number of chemical modifications are made to GNMs to facilitate the separation of anticancer drugs from GNMs at low lysosomal pH and to enable the lysosomal escape of drugs. Lysosomal escape may be associated with oxidative stress, permeabilization of the unstable membrane of cancer cell lysosomes, release of lysosomal enzymes into the cytoplasm, and cell death. GNMs can prevent or stimulate tumor cell death by inducing protective autophagy or suppressing autolysosomal degradation, respectively. Furthermore, because GNMs prevent bound fluorescent agents from emitting light, their separation in lysosomes may enable tumor cell identification and therapy monitoring. In this review, we explain how the characteristics of the lysosomal microenvironment and the unique features of tumor cell lysosomes can be exploited for GNM-based cancer therapy.

## 1. Introduction

Despite significant efforts and resources invested in cancer research, cancer is still the second leading cause of death [1]. The main problems in cancer therapy are drug resistance and the harmful side effects of drug therapy. Uncontrolled cell proliferation and a high mutation rate enable a rapid selection of drug-resistant cell clones. Chemotherapeutic agents often interfere with signaling pathways found in both malignant and healthy cells, leading to undesirable of-target effects. To reduce side effects and increase the effectiveness of cancer therapy, scientists are looking for specific characteristics of cancer cells that can be targeted [2]. Lysosomes are the membrane-enclosed digestive organelles, with a putative role in tumor growth and metastasis. Cancer cells have a large number of lysosomes, while their lysosomes have increased enzyme content and unstable membranes. These characteristics, along with the unique features of the lysosomal compartment (low pH and presence of acid hydrolases), can be effectively exploited for cancer therapy [3].

As highly biocompatible materials with unique physicochemical properties, graphene-based nanomaterials (GNMs) are widely studied for the development of innovative anticancer therapeutics. GNMs exhibit direct anti-cancer effects by inducing tumor cell death or inhibiting angiogenesis. Their selectivity against cancer cells can be improved by functionalization with targeting moieties that bind to specific receptors on tumor cells. Additionally, owing to their extensive surface area, GNMs can serve as carriers for anticancer drugs, forming graphene-based drug delivery systems (DDSs) that enhance the targeting and delivery of drugs to tumor cells. The optical properties of GNMs can be harnessed for both photothermal therapy (PTT) and real-time monitoring of cancer treatment [4,5].

GNMs are predominantly internalized in tumor cells via endocytosis, and in cases where they fail to escape from endosomes, they ultimately enter lysosomes [6,7,8]. Their lysosomal accumulation may suppress the activity of lysosomal enzymes [9], leading to a reduction in autophagic degradation, which typically serves as a protective mechanism [9,10,11], thereby contributing to tumor cell death. The sequestration of graphene-based DDSs in the lysosomal lumen prevents drugs from reaching their target site. Various functionalizations of GNMs have been developed to enable lysosomal drug escape [12,13,14,15]. Finally, GNMs can induce lysosomal membrane permeabilization (LMP) [15] and lysosomal cell death (LCD) [16,17]. This could be a promising strategy for cancer therapy due to the inherent instability of lysosomes in cancer cells.

While numerous studies in the literature have demonstrated the critical role of lysosomes in the anticancer effects of GNMs, there has been no summary review. The purpose of this article is to recapitulate the studies that have demonstrated positive and negative involvement of lysosomes in the anticancer effects of GNMs, and to explain the underlying mechanisms. To identify relevant articles, we conducted a comprehensive search of the PubMed database using the search terms “graphene,” “lysosomes,” and “cancer.”

## 2. Lysosomes in Cancer

Lysosomes are membrane-enclosed acidic organelles that contain over 60 hydrolytic enzymes, including proteases, nucleases, lipases, glycosidases, phospholipases, and phosphatases. These degradative enzymes are mainly active at the low pH (4.8) of the lysosomal lumen, which is maintained by the activity of the lysosomal proton pump V-ATPase. The main function of lysosomes is the degradation of extracellular components consumed by endocytosis, phagocytosis, or intracellular contents obtained by autophagy [18]. Lysosomes are essential for tumor growth, invasion, and metastasis. As part of the autophagy machinery, lysosomes support tumor growth by removing damaged organelles and aggregated proteins accumulated during uncontrolled cell proliferation. Autophagy also provides substrates for energy metabolism and building blocks for the de novo synthesis of macromolecules. Lysosomes promote tumor cell invasion and metastasis by secreting cathepsins and metalloproteases that degrade the extracellular matrix. Additionally, the fusion of lysosomes with the plasma membrane allows V-ATPase to pump protons from lysosomes into the extracellular space, further promoting cancer invasion. Lastly, lysosomes also capture and exocytose hydrophobic and weakly basic chemotherapeutics, leading to drug resistance [3,18].

It is not surprising that cancer cells exhibit an increase in the number, volume, and enzyme content of lysosomes. Additionally, cancer cell lysosomes are more fragile compared to normal cells, due to their larger size and unique characteristics of the lysosomal membrane structure [18]. The membrane of cancer cell lysosomes has decreased concentrations of the stabilizing molecule lysosomal-associated membrane protein (LAMP) and increased concentrations of the lysosome-destabilizing lipid sphingosine. Additionally, cancer cell lysosomes are highly sensitive to oxidative stress and enriched in iron, which is involved in the generation of highly toxic hydroxyl radicals (OH•). Consequently, cancer cell lysosomes are highly susceptible to LMP, which can lead to LCD. The enlarged and unstable lysosomes of cancer cells provide a selective advantage for tumor growth but can also represent a vulnerability that can be targeted for cancer therapy [3,18].

## 3. Graphene-Based Nanomaterials in Cancer Therapy

GNMs are materials composed of graphene or graphene-like structures, including graphene oxide, reduced graphene oxide, graphene quantum dots, and different nanocarriers made from these materials. These materials are typically between one and several carbon atoms thick and have at least one lateral dimension < 100 nm. They exhibit unique physico-chemical properties and biological behaviors [19].

Graphene is a two-dimensional material composed of a planar monolayer of carbon atoms arranged in a honeycomb lattice (Figure 1). Its high-mechanical strength and stability result from the strong sp^2^ sigma bonds that connect each atom to its three nearest neighbors. Graphene’s valence band extends across the entire sheet, giving it unusually high electron and thermal conductivity. It is highly hydrophobic, and in a hydrophilic solvent, it forms large aggregates. Additionally, graphene has an extremely high surface area (~2600 m^2^/g). Synthesis of graphene is expensive, but synthesizing other GNMs on a large scale is easier and cheaper [20,21].

Graphene oxide (GO) is a single-atom carbon layer with oxygen-containing hydrophilic functional groups, such as hydroxyl groups (-OH), epoxide groups (-O-), and carboxyl groups (-COOH) (Figure 1). These groups enable chemical modifications and high-water solubility, making it a hydrophilic alternative to graphene. GO can absorb near-infrared (NIR) radiation and convert it into heat, making it a potential candidate for PTT. Furthermore, GO exhibits excellent biocompatibility, high cellular uptake, and low toxicity [20,22,23,24].

Reduced graphene oxide (rGO) is prepared by the reduction of GO, which eliminates the oxygen-containing compounds in GO and restores the sp^2^ structure of graphene. The C/O ratio is much higher in rGO than in GO (Figure 1). Due to the loss of oxygen-containing groups, rGO exhibits hydrophobic behavior and lower dispersibility than GO. Owing to its ability to generate heat upon photoexcitation, rGO also has great potential for PTT and controlled heat-induced drug release [20,24,25].

Graphene quantum dots (GQDs) are zero-dimensional graphene nanoparticles composed of a single layer, or up to a few layers, of graphene sheets with lateral dimensions less than 100 nm (Figure 1). GQDs are chemically and physically stable, nontoxic, biocompatible, and soluble in water due to the functional groups at the edges. GQDs exhibit excellent photostability and tunable fluorescence. Therefore, they can be used for bioimaging and biosensing. The ability of GQDs to absorb radiation makes them valuable candidates for PTT and photodynamic therapy (PDT). PDT involves singlet oxygen (^1^O_2_) or other highly reactive oxygen species (ROS) produced by a light-irradiated photosensitizer to destroy abnormal cells [20,21].

Various anticancer drugs can bind to the large surface area of GNMs through π–π stacking, as well as electrostatic and hydrophobic interactions (Figure 1). Graphene-based DDSs improve drug solubility and bioavailability, protecting drugs from premature release and degradation in biological environments, thus prolonging their half-life. The functionalization of GNMs with specific antibodies or receptor ligands enhances drug concentration in the tumor tissue and stimulates its uptake in cancer cells.

However, before potentially using GNM in therapy, one should be aware of its potential negative effects on human health. While clinical studies are still pending, some animal studies have shown that GNMs are safe for clinical use, while other investigations have shown that GNMs may cause acute inflammation and chronic injury to the liver, spleen, and kidney, and may suppress embryogenesis or have other adverse effects [26]. Obviously, the toxicity of GNMs depends on their type, size, charge, functionalization, impurities, concentration, and entering route [26]. Importantly, it has been shown that the toxicity of GNMs might be reduced by their functionalization with polyethylene glycol (PEG) [27], PEGylated poly-L-lysine (PLL) [28], amine [29], and dextran groups [30].

## 4. The Transport of GNMs to Lysosomes 

When solid tumors grow larger than 2 mm in diameter, physical pressure on blood vessels reduces blood flow, resulting in a hypoxic state. To overcome this, tumor cells produce angiogenic factors to form new blood vessels. However, the newly formed blood vessels have an abnormal structure with a lack of smooth muscle layer, while pericytes surrounding the capillaries have large pores. Additionally, lymphatic drainage is limited in large tumors. Nanoscale drugs such as GNMs leak preferentially into tumor tissue through permeable tumor vessels and are then retained in the tumor due to reduced lymphatic drainage. This phenomenon is known as the enhanced permeability and retention (EPR) effect [31,32].

Next, GNMs come into contact with the surface of cancer cells, either passively or by active targeting; for example, when coupled to ligands for folate, transferrin, or epidermal growth factor receptors overexpressed on tumor cells [16,33,34], or by magnetic targeting [35]. The ability and mechanism of entry of nanoparticles into cells depend on several factors, including the type, size, and shape of nanoparticles, as well as the type of cell. Nanoparticles with a diameter between 150 nm and 3 µm can more easily pass through the plasma membrane compared to particles less than 150 nm or greater than 3 µm in diameter [36]. Cubic particles have been shown to enter the cell more efficiently than cylindrical particles [36], while hydrophilic GO enters the cell more readily than hydrophobic rGO [37]. In addition, positively charged nanoparticles pass through the negatively charged plasma membrane faster than neutral or negatively charged ones [38]. The slightly acidic condition of pH ≈ 6.8 in the tumor microenvironment, which is caused by anaerobic metabolism and accumulation of lactic acid [39], can cause negative GNMs to undergo charge reversal and become positively charged, facilitating their entry into cells [40].

Small hydrophobic graphene sheets and graphene flakes can penetrate the plasma membrane directly [41], while most other GNMs enter the cell by endocytosis. Endocytosis is an energy-dependent process by which cells take up external material by engulfing it in a vesicle derived from the cell membrane. There are different types of endocytosis, including clathrin-mediated endocytosis, caveolae-mediated endocytosis, macropinocytosis, and phagocytosis [42]. Small GO nanosheets with a diameter of 0.4 µm mainly enter cells via clathrin-mediated endocytosis, which is facilitated by cell surface receptors and results in the formation of clathrin-coated vesicles. On the other hand, large GO nanosheets with a diameter of 0.6–0.8 µm are recognized as foreign particles and are internalized through phagocytosis, where they are enclosed in an internal compartment called a phagosome [6]. GQDs cross cell membranes either via clathrin- or caveolae-mediated endocytosis [7]. The internalization of nanoparticles in caveolae-mediated endocytosis begins at the lipid rafts by the formation of caveolae vesicles [43]. Some studies suggest that rGO cannot penetrate the membrane [37], while others suggest that it is taken up by the cell through endocytosis [8] (Figure 2).

Nanoparticles that are internalized through caveolae-mediated endocytosis can be directed towards the Golgi, endoplasmic reticulum, or endosomes, forming a caveosome. This compartment usually avoids lysosomal fusion and digestion [44]. Conversely, nanoparticles that enter the cell through clathrin-coated vesicles and phagosomes are typically transported to lysosomes [45]. Clathrin-coated vesicles transport particles through early endosomes (pH 6.5), late endosomes (pH 5.5), and lysosomes (pH 4.5) [45]. Phagosomes fuse with the lysosomes, forming phagolysosomes [46]. Additionally, cytoplasmic GNMs can be captured in autophagosomes and subsequently enter autolysosomes [47,48] (Figure 2).

The degradation of GNMs in lysosomes is still uncertain. Chemical bonds between GNMs and drugs or polymers in graphene-based DDSs can be easily broken, but there is only indirect evidence of pure GNMs degrading in lysosomes. Girsh et al. demonstrated that phagocytosed graphene may be digested in macrophages, but the degradation mechanism remains unclear [49]. Additionally, Kurapati et al. showed that GO can be degraded by lysosomal enzymes neutrophil and eosinophil peroxidase in cell-free conditions [50,51]. Conversely, Xiaoli et al. demonstrated that GO cannot be digested in lysosomes. Instead, they accumulate in them, causing steric hindrance [9].

The accumulation of GNM in lysosomes may have both harmful and beneficial effects in cancer therapy. On the one hand, if anticancer drugs are attached to GNMs, their lysosomal accumulation might prevent drugs from reaching their site of action, which would reduce their anticancer effects [52]. On the other hand, it may block lysosomal activity, including protective autophagy degradation, leading to cell death and enhancing the therapeutic effect [9,10,11,53].

## 5. The Role of Lysosomes in GNM-Based Cancer Therapy

### 5.1. Lysosomes Enable Drug Release from GNM-Containing Nanocarriers, Whereupon Drugs Escape from Lysosomes

Since most chemotherapeutic agents achieve their anticancer effects by acting on cell organelles other than lysosomes (e.g., the cell nucleus, mitochondria, and endoplasmic reticulum), they must escape from the endosome/lysosome to exert their effect and avoid degradation by acid hydrolases [54]. Recently, many novel polymers, pH-sensitive peptides, proteins, and other endosomolytic agents have been synthesized to facilitate the endosomal escape [55]. Several mechanisms that are responsible for the lysosomal escape of nanoparticles could also apply to graphene-based DDSs (Figure 3). (1) The proton sponge effect is the ability of molecules, such as polyethylenimine (PEI) and polyamidoamine (PAMAM) dendrimers, that contain amine groups with buffering capacity to absorb protons, which prevents the acidification of endosomes/lysosomes. To achieve the desired pH, ATPase continues to pump protons into the endosome/lysosome, causing an influx of chloride ions and water osmosis. This process leads to an increase in pressure within the lysosomes and ultimately results in a lysosomal burst. (2) The umbrella effect is an extension of the proton sponge effect. When amines are protonated, charge repulsions cause the structure of nanoparticles to expand, leading to the rupture of endosomes/lysosomes. (3) Pore formation: Nanoparticles can fuse directly with the membrane of endosomes/lysosomes, inducing membrane stress and internal membrane tension, which can lead to the formation of pores. (4) Photochemical disruption is a mechanism in which a photosensitizer, when exposed to light, generates ROS that can destroy the integrity of the endosomal/lysosomal membrane [55,56,57].

For an anti-cancer effect, it is not necessary for the entire graphene-based DDS to leave the endosome/lysosome, only the anti-cancer drug. To achieve this, the drugs must first be separated from the GNM-containing nanocarrier. This can happen in several ways. (1) The low pH environment within lysosomes triggers the protonation of graphene-based DDSs, leading to electrostatic repulsion between positively charged drugs and nanocarriers [39,40]. (2) Low pH triggers the breakdown of the π–π interaction, the supramolecular forces that arise between unsaturated (poly)cyclic molecules with π-orbitals [58]. This has been observed for the π–π interaction between GNMs and doxorubicin [59,60,61], proflavine [61], or Chlorin e6 [62]. (3) The breakdown of chemical bonds. For example, the diselenide bond between GO and PAMAM was broken by a low lysosomal pH [15] or a high lysosomal ROS [13], resulting in the release of the drug. Next, we list studies describing how anticancer drugs are released from GNM-containing nanocarriers and the mechanisms of their endosomal/lysosomal escape (Table 1).

Wu and colleagues constructed the smart vehicle GPCP, which is composed of PLL-modified graphene conjugated with citraconine (Cit) as a charge-convertible inner layer, and PAMAM dendrimer as a cationic outer layer. GPCP was loaded with photosensitizer indocyanine green (ICG), which generates heat and ROS when irradiated in the near-infrared (NIR) and miR-21i, an oligonucleotide inhibitor of oncogenic microRNA 21. After endocytosis of the GPCP/miR-21i/ICG by MDA-MB-231 triple-negative breast cancer cells, both ICG and miR-21i were released at the acidic pH of lysosomes due to the acid-triggered charge reversal effect. Furthermore, charge-reversed GPCP destabilized the endosomal membrane through the proton sponge effect, allowing both ICG and miR-21i to escape from the endosomes/lysosomes and exert their anticancer effects. Importantly, the complex demonstrated potent antitumor activity in the NIR-irradiated MDA-MB-231 mouse model [12].

Wang and colleagues modified GO with a Poloxamer 188-modified PAMAM dendrimer, which exhibits a strong proton sponge effect. The resulting GPP complex was then loaded with the ICG. The ICG/GPP entered MCF-7 breast cancer cells by endocytosis. ROS induced the breakage of diselenide bonds between GO and PAMAM-Poloxamer 188, resulting in the release of ICG. The proton sponge effect combined with oxidative stress led to the destabilization of the lysosomal membrane, allowing for the lysosomal escape of the drug. Finally, upon NIR irradiation, GO produced heat while ICG produced both heat and ROS, leading to tumor cell death [13].

Theranostics is a term used to describe the simultaneous diagnosis and treatment of a disease. In a study by Wu et al., a graphene/gold nanostar hybrid was stabilized by PEGylation in a GO/AuNS-PEG complex. PEGylation is a widely used method to improve the hydrophilicity and biocompatibility of nanocomplexes [65]. The photosensitizer Chlorin e6 (Ce6) was loaded onto the GO/AuNS-PEG complex through π–π stacking. The resulting phototheranostic GO/AuNS-PEG/Ce6 was endocytosed into the lysosomes of EMT6 mouse breast cancer cells. Upon NIR irradiation, the lysosomal membrane was ruptured by the heat and ROS generated by GO/AuNS-PEG and Ce6, respectively. This facilitated the release of the complex from the lysosomes and its targeting of the mitochondria, ultimately inducing cell death. Treatment of EMT6 tumor-bearing mice with GO/AuNS-PEG/Ce6 and NIR irradiation enabled in-vivo photothermal imaging by an infrared camera and strongly reduced tumor growth [62].

Jahanshahi and co-workers modified fluorinated GO with two molecules with pH-dependent charge reversal properties, PEI and sericin from silkworm, to form nanocarriers named FPS. FPS was loaded with the hydrophobic natural antitumor drug curcumin. Formed FPS-Cur contained two different pH-sensitive amide linkages that were negatively charged at blood pH (≈7.4), resulting in a prolonged circulation time. At mildly acidic conditions (pH ≈ 6.5 for tumor extracellular matrix and pH ≈ 5.5 for endosomes/lysosomes), the amide linkages in FPS-Cur underwent hydrolysis, causing the complex to switch its charge from negative to positive. The induced electrostatic repulsion led to the opening of the complex’s structure, allowing for the release of curcumin. Curcumin then entered the nucleus, triggering apoptosis and necrosis of HeLa cervical cancer cells [14].

The hydrophobic antitumor drug and DNA synthesis inhibitor doxorubicin (DOX) binds to graphene and its derivatives through both hydrophobic and π–π stacking interactions. Wang et al. conjugated GO with PAMAM-Pluronic F68 (PPF68) via a diselenide bond and loaded it with DOX and ICG. The formed ICG/DOX/GO-PPF68 was internalized into multidrug-resistant (MDR) MCF-7/ADR human breast cancer cells by endocytosis, where PPF68 protected the drugs from leakage and degradation. ROS, which was generated by NIR-irradiated ICG, triggered the cleavage of the diselenide bond in the complex, resulting in the release of DOX. The proton sponge effect of PAMAM and ROS induced LMP and allowed the lysosomal escape of DOX. Upon NIR irradiation, the ROS and heat generated by ICG, the heat generated by GO, and the antiproliferative effect of DOX decreased the viability of tumor cells. Notably, the intravenously injected composite accumulated in the tumor tissues of MCF-7/ADR tumor-bearing mice due to the EPR effect and greatly reduced the tumor growth of the animals upon NIR irradiation [15].

Deng et al. loaded DOX on GO-wrapped gold nanorods, which enabled complex detection by Surface-enhanced Raman spectroscopy (SERS). At the beginning of treatment with DOX@-GO@AuNR complex, the SERS signals from GO and DOX overlapped near HeLa cells. Later, the π–π DOX/GO interaction was broken at low lysosomal pH, as evidenced by a decrease in DOX signals compared to GO SERS signals, indicating the appropriate time to irradiate the cells. The photothermal effect of NIR-irradiated GO and antiproliferative effects DOX reduced cell viability. The antitumor effect of DOX@-GO@AuNR was confirmed in NIR-irradiated HeLa-bearing mice [63].

Huang et al. developed a GO-based nanocarrier and loaded it with DOX and Ag nanoparticles as SERS substrates. The Ag-GO/DOX complex was endocytosed by Ca Ski human cervical carcinoma cells, and the low lysosomal pH disrupted the π–π interaction between DOX and GO surface, resulting in the release of DOX. SERS tracking confirmed that DOX escaped from the lysosomes into the cytoplasm and entered the nucleus, while GO remained in the cytoplasm [59].

Ma and colleagues utilized both π–π stacking and hydrophobic interactions to load DOX onto nanographene oxide (NGO). They then co-coated the formed hydrophobic NGO/DOX with a biomimetic membrane made of soy phosphatidylcholine (SPC) and modified it with a PEGylated lipid-FA (where FA stands for folic acid). The resulting NGO/DOX@SPC-FA complex demonstrated long-term blood circulation in HeLa cervical tumor-bearing mice and preferential entry into the tumor tissue due to the EPR effect. The FA moiety facilitated the binding of the complex to the FA receptor on cancer cells, followed by clathrin-dependent endocytosis. DOX was released from NGO in the acidic lysosomal environment due to the protonation effect, escaped from the lysosomes, and entered the nucleus to exert its anticancer effect [60].

Feng et al. prepared a complex of NGO, PEG, positively charged poly(allylamine hydrochloride), and 2,3-dimethylmaleic anhydride (DA) with pH-dependent charge reversibility, and then loaded it with DOX. The formed NGO-PEG-DA/DOX was negatively charged at a physiological pH of 7.4, whereas it was positively charged in the tumor microenvironment (pH = 6.8) due to the cleavage of DA. This enhanced its uptake in MCF-7 cells and provided tumor-specific cytotoxicity. At a lysosomal pH of 5, DOX was released from the complex due to its protonation-induced hydrophilicity/solubility. Moreover, electrostatic repulsion between the protonated DOX and positively charged (charge-reversed in low pH) NGO-PEG-DA also contributed to DOX release. The NGO-PEG-DA/DOX complex exhibited a synergistic anticancer effect of DOX-dependent chemotherapy and NIR-irradiated NGO-dependent photothermal therapy [40]. 

Ryu and colleagues conjugated poly(ethylene imine)-poly(l-lysine)-poly(l-glutamic acid) (PKE) with charge-conversion properties, a proton sponge polymer PEI, and rGO. The resulting PK5E7(PEI-rGO)/DOX complex was negatively charged at pH 7.4 and positively charged at pH 6. The DOX-loaded complex rapidly released the anticancer drug under lysosomal conditions due to electrostatic repulsion between the positively charged DOX and PEI-rGO. PK5E7(PEI-rGO) showed potent anticancer activity in HeLa and A549 lung adenocarcinoma cells under mildly acidic conditions mimicking the tumor microenvironment [39].

In the presence of the anticancer drug cisplatin, Nandi et al. transformed 2D sheets of GO, bound to the DNA-damaging drugs proflavin or DOX by π–π interaction, into 3D spherical GPC-NP and GDC-NP complexes, respectively. After clathrin-mediated endocytosis in HeLa cells, proflavin and DOX were released from the complexes under the influence of lysosomal acidity. Carboxylate bonds were broken, and cisplatin was also released from the complexes at low pH, returning the 3D GO to its 2D form. Ultimately, the cells died by apoptosis [61].

Wu et al. developed a mitochondria-targeting carrier (GT) by attaching the amphiphilic polymer DSPE-PEG (composed of 1,2-distearoyl-3-phosphatidylethanolamine (DSPE) as the hydrophobic tail and PEG as the hydrophilic chain) to the mitochondria-targeting moiety alkyl triphenylphosphonium (TPP) and GO. The resulting GT was then combined with the immunoconjugate diphenylmethylsilane-substituted CPG (DP-CpG) and the photosensitizer IR820. The formed GT/IR820/DP-CpG complex destabilized lysosomal membrane, escaped lysosomes, and targeted mitochondria of EMT6 mouse breast cancer cells. Upon NIR irradiation, the complex generated heat and ROS, leading to apoptosis. In EMT6 tumor-bearing mice, GT/IR820/DP-CpG produced PDT- and PTT-mediated anticancer effects and activated an anticancer immune response [64].

From previously mentioned studies, we can see that the authors mainly dealt with the separation of antitumor drugs from GNM-containing nanocarriers. The ability of acidic lysosomal milieu to induce physiochemical changes in nanocomplexes was mainly used for that purpose. With the exception of positively charged ICG/DOX/GO-PPF68 [13], all other complexes were negatively charged but became positive after protonation inside the lysosomes, which induced charge repulsion and separation of the drugs from the rest of the complexes [12,14,39,40,60]. While the ability of low lysosomal pH to provoke break downs of π–π interactions [59], amide [14], and carboxylate [61] bonds, as well as the ability ROS produced by NIR-irradiated photosensitizer to disrupt diselenide bonds, was exploited, there has been no study that clearly utilized the presence of hydrolytic enzymes for this purpose. Mechanisms of lysosomal drug escape have only been superficially addressed. While the drug escape through the lysosomal membrane, which has been damaged by the heat generated by NIR-irradiated GNMs or the combined effects of heat and ROS produced by NIR-irradiated photosensitizers, has been clearly established, [62,64], the proton sponge mechanism was more hypothesized due to the presence of the ensosomolytic molecule PAMAM than proven [12,13,15]. Moreover, lysosomal escape was stated, but its mechanism was not investigated in the rest of the above-mentioned studies [14,39,40,59,60,61]. Although the fact that antitumor drugs escaped from lysosomes seems to be more important for their antitumor activities than how they escaped from lysosomes, an elucidation of the mechanisms of escape would be a useful guide for future synthesis of graphene-based DDSs and should, therefore, be clearly elucidated.5.2. GNMs Induce Lysosomal Cell Death

Lysosomal escape may be associated with LMP. The consequence of LMP is the leakage of lysosomal enzymes into the cytoplasm. Cathepsins B, D, and L retain their activity at neutral pH, degrading cytoplasmic proteins and inducing LCD. The type of LCD induced depends on the severity of lysosomal damage; extensive LMP, with leakage of cathepsins and protons into the cytoplasm, triggers necrosis, while limited LMP triggers apoptosis. Additionally, LMP may also play a role in the initiation of autophagic cell death, necroptosis, and ferroptosis [3].

LCD may occur subsequent to the escape of nanoparticles from lysosomes via various mechanisms, including the proton sponge effect, the umbrella effect, pore formation, or photochemical disruption of the lysosomal membrane [56,66,67]. However, in some cases, LCD may not occur due to the dominance of other mechanisms of cell death. For instance, a study involving the treatment of tumor cells with ICG/DOX/GO-PPF68 detected clear LMP, but the authors attributed the antitumor effects to the antiproliferative activity of DOX, ROS generated by NIR-irradiated ICG, and hyperthermia caused by NIR-irradiated GO and ICG [15]. Furthermore, in another study, GO nanosheet-induced LMP led to lysosomal alkalinization, which inhibited autophagic degradation and, ultimately, triggered apoptosis, but not LCD [10].

It is crucial to distinguish between endosomal and lysosomal escape as endosomes have a lower acidity level and hydrolytic enzymes content. Consequently, LCD is more likely if the drug escapes from mature lysosomes rather than endosomes [56]. It is important to note that escape of anticancer drugs through pore formation may not necessarily result in the leakage of cathepsins. The formed pores may be large enough to allow for the release of the drug but not the hydrolytic enzymes. For example, Cathepsin B has a molecular weight of approximately 25,000 Da [68], while the molecular weight of DOX is only 544 Da [69].

Oxidative stress is one of the main LMP inducers [3]. Accelerated metabolism of tumor cells results in increased breakdown of iron-containing proteins and iron accumulation in lysosomes. This iron can convert ROS into highly reactive OH• via the Fenton reaction [70], which in turn disrupts lysosomal membrane integrity. Despite its short half-life, OH• is a very toxic molecule due to its high reactivity. OH• causes breaks in DNA strands, modifies DNA bases, and disrupts protein structure and function. OH• also triggers the peroxidation of lipids, which are essential components of cell membranes, including lysosomal membranes [71]. Lysosomes lack the antioxidant enzymes superoxide dismutase, catalase, and glutathione peroxidase [3], making them particularly vulnerable to oxidative stress. Accordingly, oxidative stress has been identified as the mediator of LMP in all studies involving both GNMs and LMP (Table 2).

In a study conducted by Liu et al., rat basophilic leukemia RBL2H3 cells endocytosed graphene nanosheets (GNSs), which, due to their sharp edges and rough surface, damaged the lysosomal membrane, thereby inducing LMP. GNSs also disrupted the mitochondrial electron transport chain, leading to excessive ROS production. This oxidative stress further stimulated LMP and mitochondrial membrane depolarization, ultimately resulting in apoptotic cell death [67].

Tian et al. developed a complex by assembling 1,2-distearoyl-sn-glycero-3-phosphoethanolamine-N-[folate(polyethylene glycol)-2000] (DSPE-PEG2000-FA) and a photosensitizer-labeled peptide (Ce6-Pep) on the surface of GO. This complex was internalized by HeLa cells via folate receptor-mediated endocytosis and localized in lysosomes. In lysosomes, cathepsin B cleaved the peptide, releasing Ce6 from GO. Upon red light irradiation, Ce6 induced the formation of singlet oxygen ^1^O_2_, which triggered the release of cathepsin B from lysosomes to cytosol and led to LCD [16].

Zhang and colleagues attached a PEG-modified Ru(II)-polypyridyl complex to rGO through π–π stacking and hydrophobic interactions. Upon internalization, the rGO-Ru-PEG complex was detected in lysosomes of A549 cells. Low pH and the heat produced by NIR irradiated-rGO caused the release of Ru-PEG from the rGO surface. The irradiation of cells with NIR for PTT and blue light for PDT resulted in the synergistic induction of apoptosis through the generation of ROS and the release of cathepsin B from lysosomes into the cytosol. Notably, treatment with rGO-Ru-PEG and irradiation with both NIR and blue light significantly reduced tumor growth in A549 tumor-bearing mice [17].

Liu et al. developed DHA-GO-Tf by combining the anti-malarial drug dihydroartemisinin (DHA), nanoscale GO, and transferrin (Tf), an iron transporter that can target tumor cells expressing Tf receptors. DHA-GO-Tf was endocytosed in EMT6 cells. In the acidic environment of lysosomes, iron (Fe^+3^) was released from Tf and reduced to Fe^+2^ by lysosomal ferrireductase. The reaction between DHA and Fe^2+^ generated ROS, leading to oxidative damage of lysosomes and cytotoxicity. DHA-GO-Tf was found to preferentially accumulate in the tumor tissue of EMT6-bearing mice and induce complete tumor regression [33].

In above-mentioned studies, LMP appears to be only one of the steps leading to tumor cell death. Considering the huge instability of cancer lysosomes, the ability of using LCD in GNM-based tumor therapy should be investigated in more detail. It is useful to determine whether there is a relationship between the size and charge of GNMs and their capacity to destabilize lysosomes. The ability of GNMs to induce LMP could be stimulated by their modification with amine-modified polystyrene, as this has been shown to be successful with other nanoparticles [72]. In addition, loading GNM with anticancer drugs that can themselves induce LMP, such as vincristine, vinblastine, paclitaxel, cisplatin, and doxorubicin [73], could be used for this purpose.

### 5.2. GNMs Induce Tumor Cell Death by Suppressing (Auto)Lysosomal Degradation

In the process of autophagy, exhausted/damaged intracellular macromolecules and organelles are entrapped in double-membrane autophagosomes. Autophagosomes then fuse with lysosomes in autolysosomes where sequestered content is degraded by acid hydrolases. Since autophagy is primarily responsible for degrading intracellular cytoplasmic content, it is unclear how externally added GNMs are included in autophagic vacuoles [11]. We propose several mechanisms. (1) Small nanoparticles such as graphene sheets and graphene flakes can directly diffuse through the plasma membrane into the cytoplasm [41] and subsequently become sequestered in autophagosomes. (2) Endocytosed GNMs escape from endosomes into the cytoplasm, where they can be taken up into autophagosomes, similar to what has been shown for some microorganisms [74]. (3) Late endosomes containing GNMs can fuse with autophagosomes, as already demonstrated for gold nanoparticles [75]. The main role of autophagy is to protect cells from the accumulation of damaged proteins/organelles, energy deficiency, oxidative stress, and drug-induced stress. Consistent with its pro-survival role, the induction of complete autophagy has been shown in several studies to be protective for tumor cells treated with GNMs [76,77,78,79].

On the other hand, several studies have shown that GNMs can induce tumor cell death by suppressing autophagic degradation. In the research conducted by Xiaoli and co-workers, GO induced oxidative stress, mitochondrial damage, and reduced energy production, thereby stimulating incomplete autophagy in human neuroblastoma cells SH-SY5Y. The non-degradable GO accumulated in lysosomes, causing steric hindrance that elevated lysosomal pH and suppressed the degradative capacity of lysosomal enzymes such as acid phosphatase and cathepsin B. This resulted in the accumulation of dysfunctional mitochondria and ultimately contributed to mitochondria-dependent apoptosis [9]. Feng et al. demonstrated that GO nanosheets induced LMP, impaired the acidity of lysosomes, and thus suppressed the activity of cathepsin B and acid phosphatase in PC12 rat pheochromocytoma-derived cells. The impairment of lysosomal degradation led to abnormal accumulation of the autophagic proteolysis substrate p62, which contributed in part to the induction of apoptosis [10]. Similarly, Zhang and coworkers demonstrated that GO induced lysosomal alkalization, inhibited cathepsin B, and induced p62 accumulation and apoptosis in F98 rat astroglioma cells [11]. After internalization in A549 cells via clathrin-mediated endocytosis, GO-chloroquine nanoconjugate (GO-Chl) was found to stimulate the synthesis of autophagosomes but suppress their fusion with lysosomes. This resulted in the blockade of the autophagy flux and activation of necroptotic death, as demonstrated by Arya et al. [53].

Conversely, through the selective degradation of mitochondria and cell survival proteins, prolonged and overactivated autophagy can cause apoptotic, necrotic, or necroptotic death of tumor cells, or itself be an alternative mechanism of cell death (programmed cell death Type II). Consistent with the cytotoxic role of autophagy, autophagy was induced by the combined treatment with GO and the cisplatin-stimulated necrosis of colorectal carcinoma cells [80], whereas autophagy induced by photoactivated GQDs stimulated apoptosis in glioma cells [78]. The exact mechanisms of autophagy induction/suppression by GNMs, the role of autophagy in cell death and survival, and its medical significance are summarized in our previous review [81].

Therefore, either incomplete cytoprotective autophagy [9,10,11,53] or autophagy that is itself cytotoxic [78,80] could be useful in cancer therapy with GNMs. Moreover, it might be valuable to combine widely used chemotherapeutic agents that induce pro-survival autophagy with GNMs that block autophagy flux [9,10,11,53]. On the other hand, complete pro-survival autophagy induced by GNMs [76,77,79] could be suppressed by drugs such as anti-malaric chloroquine, or anticancer drugs that have been shown to suppress autophagy flux.

GNMs were also shown to suppress lysosomal enzymes unrelated to autophagy. Yang et al. showed that PEG–GO nanocarriers increased the lysosomal accumulation of the anticancer drug tamoxifen in human hepatoma HepG2 cells. The PEG-GO/tamoxifen complex inhibited the lysosomal enzyme phospholipase, resulting in a lysosomal storage disorder called phospholipidosis [82]. This condition may be cytotoxic [83], but the sequestration of tamoxifen in lysosomes is expected to prevent its antitumor effects mediated by a blockade of estrogen receptors [82]. However, the effect of nanocarriers on the anticancer activity of the drug still requires investigation.

### 5.3. Lysosomes Enable Detection of Cancer Cells by the GNMs

Finally, lysosomes are not only directly involved in the antitumor activity of GNMs, but the specificities of the lysosomal environment, such as its acidity or the presence of proteolytic enzymes, may also facilitate the identification of tumor cells, as demonstrated in the following studies.

Mosaiab and colleagues loaded DOX onto a (CA-BDP)-PPDN/rGO complex, composed of chloro-3′,4′-dihydroxyacetophenone (CA), a fluorescent dye boron-dipyrromethane [BODIPY(BDP)], PEGylated pH-sensitive *N*,*N*-dimethylacrylamide and thermosensitive N-isopropylacrylamide (PPDN), and rGO. Within the nanocomposite, rGO quenched the emission of [BODIPY(BDP)] fluorescence at a normal pH of 7.4, but not at a pH of 6, which mimicked a tumor microenvironment, or a pH of 5, which mimicked lysosomes. Accordingly, the complex displayed intense fluorescence in the lysosomes of human MDA-MB 231 breast cancer cells. Therefore, the complex has potential not only as an antitumor therapy mediated by DOX, but also as a fluorescent probe to identify cancer cells [84].

In the aforementioned research by Tian et al., GO was used to quench the fluorescence of the photosensitizer Ce6 within a nanocomposite composed of Ce6-Pep, DSPE-PEG2000-FA, and GO. Upon entering the lysosomes of HeLa cells, cathepsin B cleaved the peptide to release the fluorescent Ce6 from GO. The lysosomal pH also stimulated the discharge of Ce6 from the nanocomposite. The nanocomposite was internalized by cancer cells in mice bearing HeLa tumors via folate receptor-mediated endocytosis and displayed strong fluorescence after entering their lysosomes, enabling the identification of cancer cells in vivo [16].

Finally, in the previously mentioned study by Wu and colleagues, graphene from the smart vehicle GPCP quenched the fluorescence of the photosensitizer ICG. When ICG was released in lysosomes due to the acid-induced charge reversal effect, it began to fluoresce. Since the complex preferentially accumulated in tumor tissue, fluorescent tumors were readily observed in the MDA-MB-231 tumor-bearing mice [12].

It is worth mentioning that some studies discussed earlier have used GNMs for imaging purposes. However, we avoided discussing them in this section for two reasons: the lysosomal degradation of graphene-based DDSs did not permit specific tumor cell detection but only intracellular localization of the DDSs [59,63]; and GNM-dependent imaging of tumor cells was not related to lysosomal activity [62]. Fluorescence not only plays a crucial role in the detection of tumor cells but can also be important in determining the exact time and location of irradiation in PTT and PDT.

## 6. Conclusions and Future Perspectives

Most studies that mention lysosomes in connection with GNN-based anticancer therapy, focus on the release of anticancer drugs from GNM-containing nanocarriers within lysosomes and their subsequent lysosomal escape. The release of drugs from the nanocarriers was mainly triggered by the physicochemical changes of the nanocarriers and drugs in response to low pH. As only one study has reported lysosomal enzyme-induced drug release, this approach may have potential for further exploration. Moreover, numerous studies showed that GNMs affect cancer cell death by modulating autophagy flux. However, only a few studies have shown that GNMs can stimulate LMP and LCD. ROS generated by either GNMs or GNM-bound drugs (light-irradiated photosensitizers) were mediators of LMP in all cases. Considering that lysosomes lack antioxidant defenses and are rich in iron, and that lysosomes of cancer cells are extremely unstable, LCD could be an effective approach to treat cancer cells, especially those that are apoptosis-resistant [3]. Thus, a promising avenue for future research may be to enhance the ability of GNM to induce LMP by chemical modifications, as has been shown with other types of nanoparticles [78], or by loading GNM-based nanocarriers with LMP-inducing drugs, which include some of the commonly used chemotherapeutics. Since GQDs were shown to enter the lysosomes [80] and produce ROS when exposed to light, their ability to induce LMP/LCD should be investigated. It is important to be aware of the limitations of the potential clinical use of GNMs, as there are no clinical studies showing that their use in humans is safe. Of concern is their demonstrated toxicity in animal models. However, since it has been shown that there are procedures that reduce the toxicity of GNMs, it is possible to envision the future development of GNMs that are safe for therapeutic applications.

## Figures and Tables

**Figure 1 pharmaceutics-15-01846-f001:**
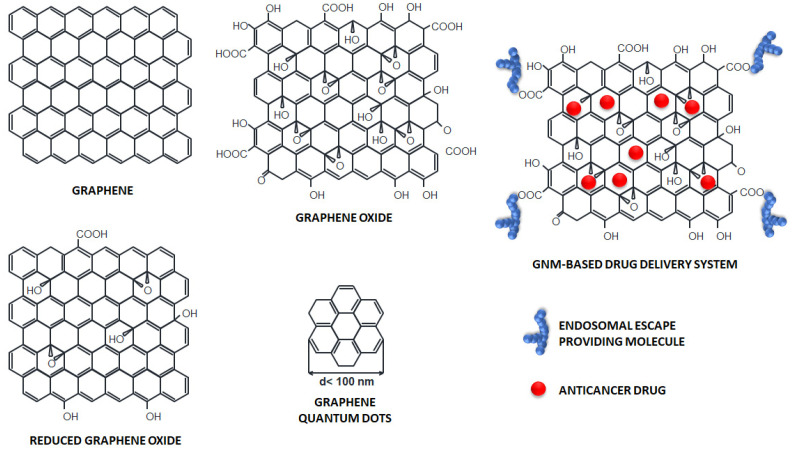
The structure of GNMs and graphene-based DDSs.

**Figure 2 pharmaceutics-15-01846-f002:**
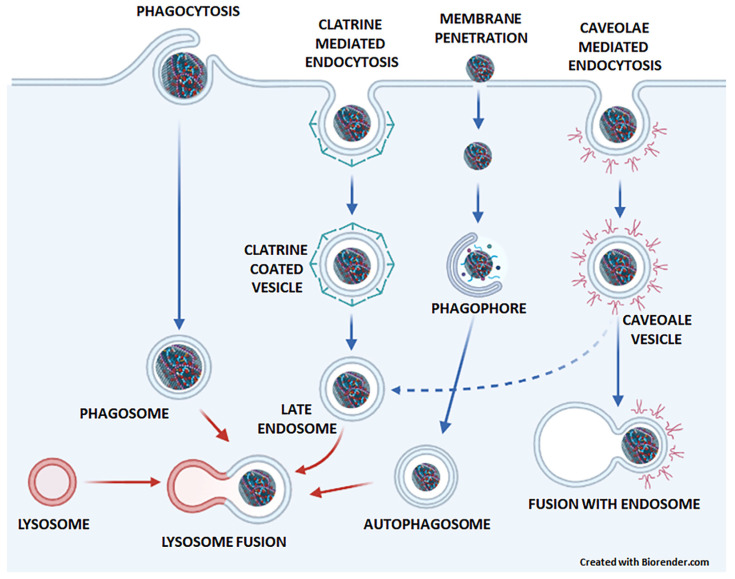
The entry of GNMs and graphene-based DDSs into cells. In phagocytosis, large GO are engulfed in a phagosome, which fuses with the lysosome. In clathrin-mediated endocytosis, small GO nanosheets and GQDs are bound to cell surface receptors and internalized into clathrin-coated vesicles that mature into late endosomes and eventually fuse with the lysosome. Small hydrophobic graphene sheets and flakes enter the cytoplasm directly by penetrating the plasma membrane and can become entrapped in autophagosomes, which subsequently fuse with lysosomes in the process of autophagy. GQDs can also cross cell membranes by caveolae-mediated endocytosis, which occurs through the formation of caveolae vesicles, which may fuse with endosomes.

**Figure 3 pharmaceutics-15-01846-f003:**
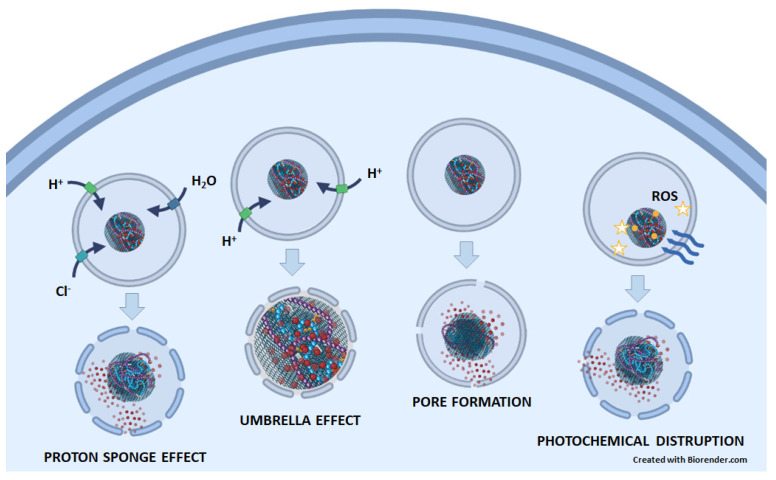
The possible mechanisms of endosomal/lysosomal escape of anticancer drugs (red circles) released from GNMs. In the proton sponge effect, GNM-bound molecules containing amine groups with buffering capacity absorb protons. Therefore, the ATPase continuously pumps protons into the endosome/lysosome, which is followed by an influx of chloride ions, water osmosis, increased pressure, and, finally, lysosomal rupture. In the umbrella effect, protonation of amines leads also to charge repulsion and nanoparticle expansion, which causes endosomal/lysosomal rupture. By inducing membrane stress and internal membrane tension, nanoparticles can lead to the formation of pores on the endosome/lysosome membrane. In photochemical disruption, a light-irradiated (blue wavy arrows) photosensitizer (yellow circles) attached to a GNM-containing nanocarrier generates ROS (yellow stars), which destroys the integrity of the endosomal/lysosomal membrane.

**Table 1 pharmaceutics-15-01846-t001:** Lysosomes enable drug release from GNMs, which is followed by endosomal/lysosomal escape of the drug. → denotes causal connection; ↑ denotes increase; ? denotes unknown mechanism. Abbreviations for graphene-based DDSs in order of appearance: GPCP/miR-21i/ICG, poly(l-lysine)-modified graphene conjugated with citraconine and polyamidoamine loaded with indocyanine green and miR-21i; ICG/GPP, graphene oxide modified with poloxamer 188-modified polyamidoamine-dendrimer; GO/AuNS-PEG/Ce6, PEGylated graphene/gold nanostar hybrid loaded with Chlorin e6; FPS, fluorinated GO modified with PEI and sericin; FPS-Cur, FPS loaded with curcumin; ICG/DOX/GO-PPF68, graphene oxide conjugated with polyamidoamine-pluronic F68 loaded with doxorubicin and indocyanine green; DOX@-GO@AuNR, gold nanorods wrapped in graphene oxide and loaded with doxorubicin; Ag-GO/DOX, graphene oxide loaded with doxorubicin and Ag; NGO/DOX@SPC-FA, nanographene oxide loaded with doxorubicin, co-coated with soy phosphatidylcholine and modified with a PEGylated folic acid; NGO-PEG-DA/DOX, nanographene oxide conjugated with polyethylene glycol and poly(allylamine hydrochloride), modified with 2,3-dimethylmaleic anhydride and loaded with doxorubicin; PK5E7(PEI-rGO), poly(ethylene imine) coated with poly(ethylene imine)-poly(l-lysine)-poly(l-glutamic acid) and conjugated with reduced graphene oxide; GPC-NP, graphene oxide transformed into 3D in the presence of cisplatin and loaded with proflavin; GDC-NP, graphene oxide transformed into 3D in the presence of cisplatin and loaded with doxorubicin; GT/IR820/DP-CpG, 1,2-distearoyl-3-phosphatidylethanolamine and polyethylene glycol conjugated with alkyl triphenylphosphonium and graphene oxide, loaded with diphenylmethylsilane-substituted CPG and photosensitizer IR820.

Graphene-Based DDSs	Cells	Drug Detachment Mechanism	Escape Mechanism	Ref
GPCP/miR-21i/ICG	MDA-MB-231	Low pH → charge conversion of PLL-Cit from positive to negative → conformational changes of PAMAM → cleavage ICG/graphene interaction	Proton sponge effect	[12]
ICG/GPP	MCF-7	ROS → cleavage of diselenide bond between GO and PAMAM-Poloxamer 188 → ICG release	Proton sponge effect	[13]
GO/AuNS-PEG/Ce6	EMT6	?	Heat and ROS disrupted lysosomal membrane	[62]
FPS-Cur	HeLa	Hydrolysis of amide links + charge reversion to positive FPS → electrostatic repulsion → structure opening	?	[14]
ICG/DOX/GO-PPF68	MCF-7/ADR	NIR → ICG → ROS → cleavage of diselenide bond	Proton sponge effect	[15]
DOX@-GO@AuNR	HeLa	Low pH → DOX release	?	[63]
Ag-GO/DOX	Ca Ski	Low pH → breakdown of π-π DOX/GO interaction	?	[59]
NGO/DOX@SPC-FA	HeLa	Low pH → protonation effect → DOX release	?	[60]
NGO-PEG-DA/DOX	MCF-7	Low pH → DOX protonation → ↑ DOX hydrophilicity/solubility + electrostatic repulsion with charge reversion to positive NGO-PEG-DA	?	[40]
PK5E7(PEI-rGO)	HeLa A549	Low pH → DOX protonation → electrostatic repulsion with charge reversion to positive PEI-rGO	?	[39]
GPC-NP and GDC-NP	HeLa	Low pH → DOX and proflavine release + cleavage of carboxylate bonds and cisplatin release → GO converts from 3D into 2D	?	[61]
GT/IR820/DP-CpG	EMT6	?	Heat and ROS disrupted lysosomal membrane	[64]

**Table 2 pharmaceutics-15-01846-t002:** GNMs and graphene-based DDSs induce LMP. → denotes causal connection; ↓ denotes decrease/inhibition. ROS—reactive oxygen species. Abbreviations for GNMs/graphen-based DDSs in order of appearance: GNSs, graphene nanosheets; DSPE-PEG2000-FA, 1,2-distearoyl-sn-glycero-3-phosphoethanolamine-N-[folate(polyethylene glycol)-2000]; Ce6, Chlorin e6; GO, graphene oxide; rGO-Ru-PEG, reduced graphene oxide loaded with Ru(II)-polypyridyl complex modified with polyethylene glycol; DHA, dihydroartemisinin; Tf, transferrin.

GNMs/Graphene-Based DDSs	Cell Type	Mechanism of LMP Induction	Ref
GNSs	RBL2H3	Sharp edges and rough surface of GNSs + ↓ mitochondrial electron transport chain → ROS	[67]
DSPE-PEG2000-FA + Ce6-Pep + GO	HeLa	Light → Ce6 → ^1^O_2_ → cathepsin B release	[16]
rGO-Ru-PEG	A549	Light → Ru → ROS → cathepsin B release	[17]
DHA-GO-Tf	EMT6	Low pH → release of Fe^+3^ from the Tf → Fe^+3^ reduced to Fe^+2^ → Fe^+2^ + DHA → ROS → lysosomal damage	[33]

## Data Availability

Not applicable.

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
