# Peer review of "The Exploitation of Lysosomes in Cancer Therapy with Graphene-Based Nanomaterials"

_pharmaceutics, 2023, doi:10.3390/pharmaceutics15071846_

Round 1

Reviewer 1 Report

The manuscript "The exploitation of lysosomes in cancer therapy with graphene-based nanomaterials" explain how the characteristics of the lysosomal microenvironment and the unique features of tumor cell lysosomes can be exploited for GNM-based cancer therapy. The review is well written, but before its publication some issues have to be addressed:

- could the authors explain in more detail how OH are "highly toxic hydroxyl radicals" and add the information to the manuscript;

- how low the toxicity of graphene-based nanomaterials could be? In the literature are many contradictory studies. Maybe the authors should stress a little bit more on this aspect and add some data regarding the toxicity of graphene-based nanomaterials (https://particleandfibretoxicology.biomedcentral.com/articles/10.1186/s12989-016-0168-y/tables/1).

- what supports the following affirmation "The accumulation of GNM in lysosomes may have both harmful and beneficial effects in cancer therapy. On the one hand, it may prevent drugs from reaching their site of action, reducing their anticancer effects. On the other hand, it may block lysosomal activity, leading to cell death, thus enhancing the therapeutic effect."? I would expect to find out any of the information in the literature elsewhere. Please cite references supporting that.

- nowhere in the manuscript could be found what FPS does mean. The authors have to check carefully each abbreviation and give their meaning.

- the source for DOX molecular weight should be listed in the Reference section.

- conclusions have to not contain bibliographic references. The authors have to remove them.

The English language is readable and understandable.

Author Response

Dear reviewer,

Thank you for your useful suggestions. We hope that we have fulfilled everything you suggested.

- could the authors explain in more detail how OH are "highly toxic hydroxyl radicals" and add the information to the manuscript;

  1. In accordance with the reviewer's suggestion, we have added the following statement on the toxicity of hydroxyl radicals on page 12:

Despite its short half-life, OH· is a very toxic molecule due to its high reactivity. OH· causes breaks in DNA strands, modifies DNA bases, disrupts protein structure and function. OH· also triggers the peroxidation of lipids, which are essential components of cell membranes, including lysosomal membranes (10.3390/ijms22094642).

- how low the toxicity of graphene-based nanomaterials could be? In the literature are many contradictory studies. Maybe the authors should stress a little bit more on this aspect and add some data regarding the toxicity of graphene-based nanomaterials  

  1. The reviewer is right. This is a very important aspect of GNMs that we did not describe in the first version of the paper, so we have added it now on page 4,

However, before potentially using GNM in therapy, one should be aware of its potential negative effects on human health. While clinical studies are still pending, some animal studies have shown that GNMs are safe for clinical use, while other investigations have shown that GNMs may cause acute inflammation and chronic injury to the liver, spleen, and kidney, and may suppress embryogenesis or have other adverse effects (10.1186/s12989-016-0168-y). Obviously, the toxicity of GNMs depends on their type, size, charge, functionalization, impurities, concentration, and entering route (10.1186/s12989-016-0168-y). Importantly, it has been shown that the toxicity of GNMs might be reduced by their functionalization with PEG (10.1021/nn1024303), PEGylated poly-L-lysine (PLL)(10.1002/adma.201202678), amine (10.1021/nn300172t), and dextran groups (10.1016/j.carbon.2011.05.056).

as well in section 6: Conclusions and future perspectives, on page 16:

It is important to be aware of the limitations of the potential clinical use of GNMs, as there are no clinical studies showing that its use in humans is safe. Of concern is their demonstrated toxicity in animal models. However, since it has been shown that there are procedures that reduce the toxicity of GNMs, it is possible to envision the future development of GNMs that are safe for therapeutic applications.

- what supports the following affirmation "The accumulation of GNM in lysosomes may have both harmful and beneficial effects in cancer therapy. On the one hand, it may prevent drugs from reaching their site of action, reducing their anticancer effects. On the other hand, it may block lysosomal activity, leading to cell death, thus enhancing the therapeutic effect."? I would expect to find out any of the information in the literature elsewhere. Please cite references supporting that.

  1. We have supported of our claims by incorporating relevant references (page 6):

The accumulation of GNMs in lysosomes may have both harmful and beneficial effects in cancer therapy. On the one hand, if anticancer drugs are attached to GNMs, their lysosomal accumulation might prevent drugs from reaching their site of action, which would reduce their anticancer effects (10.3390/ijms21124392). On the other hand, it may block lysosomal activity, including protective autophagy degradation, leading to cell death, thus enhancing the therapeutic effect (10.1016/j.jhazmat.2021.126158; 10.1016/j.actbio.2018.09.057; 10.1186/s12951-020-00605-6; 10.2217/nnm-2018-0086).

- nowhere in the manuscript could be found what FPS does mean. The authors have to check carefully each abbreviation and give their meaning.

  1. We have carefully checked all the abbreviations.

Fluorinated GO modified with PEI and sericin was referred to as FPS by Jahanshahi et al (10.1016/j.ijpharm.2019.118791).

We have added meaning of FPS and FPS-Cur in Table 1 legend (page 8):

FPS, fluorinated GO modified with PEI and sericin; FPS-Cur, FPS loaded with curcumin

and explained it in text (page10):

Jahanshahi and co-workers modified fluorinated GO with two molecules with pH-dependent charge reversal properties PEI and sericin from silkworm to form nanocarrier named FPS. FPS was loaded with the hydrophobic natural antitumor drug curcumin.

- the source for DOX molecular weight should be listed in the Reference section.

  1. We have added references for the molecular weights of both cathepsin B and DOX (page 12):

For example, cathepsin B has a molecular weight of approximately 25,000 Da (10.1016/B978-0-12-384731-7.00248-8), while the molecular weight of DOX is only 544 Da (10.1038/sj.neo.7900096).

- conclusions have to not contain bibliographic references. The authors have to remove them.

  1. We have removed all the references from section 6. Conclusions and future perspectives (page 16).

Reviewer 2 Report

The manuscript "The exploitation of lysosomes in cancer therapy with graphene-based nanomaterials", written by Ristic et al., is well written and aims at a very interesting topics of drug delivery systems based on graphene nanoplatforms. The present study describes interactions of graphene with lysosomes. The structure and chapters of the work are well planned, authors cite recent publications and the text contains data based on recent findings. I have only a few minor comments:

1. chapters 2 and 3 could be shorter. They serve as an introduction to the theme, and more details on GNPs and lysosomes can be found in other papers. 

2. Introduction of SERS is not necessary and moves the attention away from the core topic of the manuscript.

3. Most of the text in the chapter 5 only describes other works without any provided comments from the authors. The whole chapter should also contain a critical overview/insight, provided by authors, of the referenced manuscripts.

Author Response

Dear reviewer,

Thank you for your helpful suggestions. We hope that we have fulfilled your requirements.

  1. chapters 2 and 3 could be shorter. They serve as an introduction to the theme, and more details on GNPs and lysosomes can be found in other papers.

The reviewer is correct that the facts mentioned in Chapters 2 and 3 can be found in many other reviews, but we still wanted to inform or remind readers of basic facts before moving on to the more challenging parts of the text. Chapter 2 is already very concise, and we have not shortened it further. However, there was room to shorten chapter 3, and we have deleted a lot of parts that are not crucial for someone who is not fully familiar with the subject to understand the rest of the text. Please see pages 3 and 4.

  1. Introduction of SERS is not necessary and moves the attention away from the core topic of the manuscript.

In order not to move attention away from the main topic, we have omitted the explanation of SERS and summarized it in one sentence on page 10:

Deng et al. loaded DOX on GO-wrapped gold nanorods, which enabled complex detection by Surface-enhanced Raman spectroscopy (SERS).

  1. Most of the text in the chapter 5 only describes other works without any provided comments from the authors. The whole chapter should also contain a critical overview/insight, provided by authors, of the referenced manuscripts.

To solve this problem, we have added critical and concluding paragraphs for all the chapters 5.1-5.4.

5.1. (Pages 11 and 12): From previously mentioned studies we can see that the authors mainly dealt with separation of antitumor drugs from GNM-containing nanocariers. The ability of acidic lysosomal milieu to induce physiochemical changes in nanocomplexes was mainly used for that purpose. With exception of positively charged ICG/DOX/GO-PPF68 (10.1016/j.ejps.2019.04.021), all other complexes were negatively charged but become positive after protonation inside the lysosomes, which induced charge repulsion and separation of the drugs from rest of the complexes (10.1002/adhm.201901187; 10.1016/j.ijpharm.2019.11879; 10.1039/c8tb00804c; 10.1002/adhm.201300549; 10.3390/nano9091289). While ability of low lysosomal pH to provoke break-down of π-π interactions (10.1039/c3nr03264g), amide (10.1016/j.ijpharm.2019.118791) and carboxylate (10.1039/c6cc09006k) bonds, as well as ability ROS produced by NIR-irradiated photosensitizer to disrupt diselenide bonds were exploited, there has been no study that clearly utilized the presence of hydrolytic enzymes for this purpose. Mechanisms of lysosomal drug escape have only been superficially addressed. While drug escape through lysosomal membrane damaged by heat produced by NIR-irradiated GNMs and heat and ROS produced by photosensitizers was established (10.1016/j.actbio.2017.01.078; 10.1016/j.actbio.2017.01.078), the proton sponge mechanism was more hypothesized due to the presence of the ensosomolytic molecule PAMAM than proven (10.1016/j.actbio.2017.01.078; 10.1016/j.actbio.2017.01.078; 10.1016/j.actbio.2017.01.078). Moreover, lysosomal escape was stated but its mechanism was not investigated in rest of above-mentioned studies (10.1016/j.ijpharm.2019.118791; 10.1039/c3nr03264g; 10.1039/c8tb00804c;  10.1002/adhm.201300549; 10.3390/nano9091289; 10.1039/c6cc09006k). Although the fact that antitumor drugs escaped from lysosomes seems to be more important for their antitumor activities than how they escaped from lysosomes, elucidation of the mechanisms of escape would be a useful guide for future synthesis of graphene-based DDSs and should therefore be clearly elucidated.

5.2. (Pages 13 and 14): In above mentioned studies LMP appears to be only one of steps leading to tumor cell death. Considering huge instability of cancer lysosomes, the ability of using LCD in GNM-based tumor therapy should be investigated in more details. It is useful to determine whether there is a relationship between the size and charge of GNMs and their capacity to destabilize lysosomes. The ability of GNMs to induce LMP could be stimulated by their modification with amine-modified polystyrene, as this has been shown to be successful with other nanoparticles (10.1098/rsob.170271). In addition, loading GNM with anticancer drugs that can themselves induce LMP, such as vincristine, vinblastine, paclitaxel, cisplatin, and doxorubicin (10.3390/ijms24032176), could be used for this purpose.

5.3. (Pages 14 and 15): Therefore, either incomplete cytoprotective autophagy (10.1016/j.jhazmat.2021.126158; 10.1016/j.actbio.2018.09.057; 10.1186/s12951-020-00605-6; 10.2217/nnm-2018-0086) or autophagy that is itself cytotoxic (10.7150/thno.24173; 10.1016/j.biomaterials.2012.06.060) could be useful in cancer therapy with GNMs. Moreover, it might be valuable to combine widely used chemotherapeutic agents that induce prosurvival autophagy with GNMs that block autophagy flux (10.1016/j.jhazmat.2021.126158; 10.1016/j.actbio.2018.09.057; 10.1186/s12951-020-00605-6; 10.1016/j.biochi.2019.02.012). On the other hand, complete prosurvival autophagy induced by GNMs (10.1016/j.freeradbiomed.2021.10.025; 10.1016/j.biomaterials.2014.11.034; 10.1016/j.jes.2018.07) could be suppressed by drugs such as antimalaric chloroquine or anticancer drugs which suppress autophagy flux.

5.4. (Page15): Fluorescence not only plays a crucial role in the detection of tumor cells but can also be important in determining the exact time and location of irradiation in PTT and PDT.

Best regards,

the authors

Reviewer 3 Report

The article by Ristic et al. gives a brief review of GNM-based cancer therapy. The article is well organized. In my opinion, a minor revision should be performed before publication.

1)      I would like to recommend the authors give the number of papers in this field in order to justify a review on this topic.

2)      Review should be critical instead of plain text.

3)      The limitation and challenges should be covered. The last section should be the conclusion, challenges, and future perspectives.

Author Response

Dear reviewer,

Thank you very much. We hope that we have improved the work according to your suggestions.

1)     I would like to recommend the authors give the number of papers in this field in order to justify a review on this topic.

These are our papers related to the subject of review and they are listed among other references (we do not know where else to mention the number of works in this field to justify a review on this topic):

  1. M. Krunić, B. Ristić, M. Bošnjak, V. Paunović, G. Tovilović-Kovačević, N. Zogović, A. Mirčić, Z. Marković, B. Todorović-Marković, S. Jovanović, D. Kleut, M. Mojović, Đ. Nakarada, O. Marković, I. Vuković, L. Harhaji-Trajković, V. Trajković, Graphene quantum dot antioxidant and proautophagic actions protect SH-SY5Y neuroblastoma cells from oxidative stress-mediated apoptotic death, Free Radical Biology and Medicine, 177 (2021) 167-180.
  2. B. Ristic, L. Harhaji-Trajkovic, M. Bosnjak, I. Dakic, S. Mijatovic, V. Trajkovic, Modulation of Cancer Cell Autophagic Responses by Graphene-Based Nanomaterials: Molecular Mechanisms and Therapeutic Implications, Cancers, 13 (2021).
  3. V. Paunovic, M. Kosic, M. Misirkic-Marjanovic, V. Trajkovic, L. Harhaji-Trajkovic, Dual targeting of tumor cell energy metabolism and lysosomes as an anticancer strategy, Biochimica et biophysica acta. Molecular cell research, 1868 (2021) 118944.
  4. Z.M. Markovic, B.Z. Ristic, K.M. Arsikin, D.G. Klisic, L.M. Harhaji-Trajkovic, B.M. Todorovic-Markovic, D.P. Kepic, T.K. Kravic-Stevovic, S.P. Jovanovic, M.M. Milenkovic, D.D. Milivojevic, V.Z. Bumbasirevic, M.D. Dramicanin, V.S. Trajkovic, Graphene quantum dots as autophagy-inducing photodynamic agents, Biomaterials, 33 (2012) 7084-7092.

2)      Review should be critical instead of plain text.

We fully agree with this objection. We have added some criticisms, suggestions and conclusions for all the chapters 5.1-5.4.

5.1. (Pages 11 and 12): From previously mentioned studies we can see that the authors mainly dealt with separation of antitumor drugs from GNM-containing nanocariers. The ability of acidic lysosomal milieu to induce physiochemical changes in nanocomplexes was mainly used for that purpose. With exception of positively charged ICG/DOX/GO-PPF68 (10.1016/j.ejps.2019.04.021), all other complexes were negatively charged but become positive after protonation inside the lysosomes, which induced charge repulsion and separation of the drugs from rest of the complexes (10.1002/adhm.201901187; 10.1016/j.ijpharm.2019.11879; 10.1039/c8tb00804c; 10.1002/adhm.201300549; 10.3390/nano9091289). While ability of low lysosomal pH to provoke break-down of π-π interactions (10.1039/c3nr03264g), amide (10.1016/j.ijpharm.2019.118791) and carboxylate (10.1039/c6cc09006k) bonds, as well as ability ROS produced by NIR-irradiated photosensitizer to disrupt diselenide bonds were exploited, there has been no study that clearly utilized the presence of hydrolytic enzymes for this purpose. Mechanisms of lysosomal drug escape have only been superficially addressed. While drug escape through lysosomal membrane damaged by heat produced by NIR-irradiated GNMs and heat and ROS produced by photosensitizers was established (10.1016/j.actbio.2017.01.078; 10.1016/j.actbio.2017.01.078), the proton sponge mechanism was more hypothesized due to the presence of the ensosomolytic molecule PAMAM than proven (10.1016/j.actbio.2017.01.078; 10.1016/j.actbio.2017.01.078; 10.1016/j.actbio.2017.01.078). Moreover, lysosomal escape was stated but its mechanism was not investigated in rest of above-mentioned studies (10.1016/j.ijpharm.2019.118791; 10.1039/c3nr03264g; 10.1039/c8tb00804c;   10.1002/adhm.201300549; 10.3390/nano9091289; 10.1039/c6cc09006k). Although the fact that antitumor drugs escaped from lysosomes seems to be more important for their antitumor activities than how they escaped from lysosomes, elucidation of the mechanisms of escape would be a useful guide for future synthesis of graphene-based DDSs and should therefore be clearly elucidated.

5.2. (Pages 13 and 14): In above mentioned studies LMP appears to be only one of steps leading to tumor cell death. Considering huge instability of cancer lysosomes, the ability of using LCD in GNM-based tumor therapy should be investigated in more details. It is useful to determine whether there is a relationship between the size and charge of GNMs and their capacity to destabilize lysosomes. The ability of GNMs to induce LMP could be stimulated by their modification with amine-modified polystyrene, as this has been shown to be successful with other nanoparticles (10.1098/rsob.170271). In addition, loading GNM with anticancer drugs that can themselves induce LMP, such as vincristine, vinblastine, paclitaxel, cisplatin, and doxorubicin (10.3390/ijms24032176), could be used for this purpose.

5.3. (Pages 14 and 15): Therefore, either incomplete cytoprotective autophagy (10.1016/j.jhazmat.2021.126158; 10.1016/j.actbio.2018.09.057; 10.1186/s12951-020-00605-6; 10.2217/nnm-2018-0086) or autophagy that is itself cytotoxic (10.7150/thno.24173; 10.1016/j.biomaterials.2012.06.060) could be useful in cancer therapy with GNMs. Moreover, it might be valuable to combine widely used chemotherapeutic agents that induce prosurvival autophagy with GNMs that block autophagy flux (10.1016/j.jhazmat.2021.126158; 10.1016/j.actbio.2018.09.057; 10.1186/s12951-020-00605-6; 10.1016/j.biochi.2019.02.012). On the other hand, complete prosurvival autophagy induced by GNMs (10.1016/j.freeradbiomed.2021.10.025; 10.1016/j.biomaterials.2014.11.034; 10.1016/j.jes.2018.07) could be suppressed by drugs such as antimalaric chloroquine or anticancer drugs which suppress autophagy flux.

5.4. (Page15): Fluorescence not only plays a crucial role in the detection of tumor cells but can also be important in determining the exact time and location of irradiation in PTT and PDT.

3)      The limitation and challenges should be covered. The last section should be the conclusion, challenges, and future perspectives.

The reviewer is right. To address this issue, we added the text at the end of the article (page 16).

It is important to be aware of the limitations of the potential clinical use of GNMs, as there are no clinical studies showing that its use in humans is safe. Of concern is their demonstrated toxicity in animal models. However, since it has been shown that there are procedures that reduce the toxicity of GNMs, it is possible to envision the future development of GNMs that are safe for therapeutic applications.